



# A new multi-resolution bathymetric dataset of the Gulf of Naples (Italy) from complementary multi-beam echosounders

Federica Foglini[1], Marzia Rovere[1], Renato Tonielli[1], Giorgio Castellan[1,2*], Mariacristina Prampolini[1,2], Francesca Budillon[1], Marco Cuffaro[3], Gabriella Di Martino[1], Valentina Grande[1], Sara Innangi[1], Maria Filomena Loreto[1], Leonardo Langone[4], Fantina Madricardo[1], Alessandra Mercorella[1], Paolo Montagna[4], Camilla Palmiotto[1], Claudio Pellegrini[1], Antonio Petrizzo[1], Lorenzo Petracchini[3], Alessandro Remia[1], Marco Sacchi[1], Daphnie Sanchez Galvez[1], Anna Nora Tassetti[5], Fabio Trincardi[6].

[1] CNR –ISMAR - National Research Council, Institute of Marine Sciences, Italy;
[2] NBFC - National Biodiversity Future Centre, Italy
[3] CNR-IGAG - National Research Council, Institute of Environmental Geology and Geoengineering, Italy;
[4] CNR ISP - National Research Council, Institute of Polar Sciences, Italy;
[5] CNR IRBIM- National Research Council, Institute for Biological Resources and Marine Biotechnologies, Italy;
[6] CNR-DSSTTA - National Research Council, Department of Earth Systems Science and Environmental Technologies, Italy.

*Correspondence to*: giorgio.castellan@cnr.it

## Abstract

High-resolution bathymetry provides critical information to marine geoscientists. Bathymetric big data help characterise the seafloor and its benthic habitats, understand sedimentary records, and support the development of offshore engineering infrastructures. From September 27th to October 20th, 2022, the new CNR Research Vessel GAIA BLU explored the seafloor of the Naples and Pozzuoli Gulfs, and the Amalfi coastal area (Tyrrhenian Sea, Italy) from 50 to more than 2000 m water depth, acquiring about 5000 km$^2$ of multi beam echosounder data. This area is particularly vulnerable to abrupt changes driven by the dynamics of several volcanic complexes, active in the area, and by human-induced impacts reflecting the proximity to the highly populated and touristic coastal area of Naples and nearby famous islands. For these reasons, the seafloor of the area needs to be known and constantly monitored. The digital bathymetric data previously available are restricted to the shallow highly dynamic area of the Gulf of Naples and appear fragmented as they were acquired in successive years, with different goals thereby using a variety of devices, with markedly different spatial resolutions. In this paper, we present bathymetric maps of the Gulf of Naples and adjacent slope basins at unprecedented resolution using three state-of-the-art multi beam echosounders. These high-resolution data highlight the technological advances of geophysical surveys achieved over the last 20 years and contribute to assessing the most dynamic areas where changes in the seafloor over time can be quantified. The new digital multi-resolution bathymetric products are openly accessible via Marine Geosciences Data System MGDS (refer to





section Data Availability, Table 8, for datasets and products DOIs), perfectly matching the FAIR (Findable, Accessible,
Interoperable and Reusable) and Open Science Principles.

## 1. Introduction

In 2018, GEBCO and the Nippon Foundation joined forces to establish the Nippon Foundation GEBCO Seabed 2030 Project
(Mayer et al., 2018), an international effort to foster the complete mapping of the world ocean by 2030. Despite many years
of mapping efforts unveiled increasingly larger portions of the seabed, only about 25% of the world oceans seafloor is mapped
to date at high resolution (https://seabed2030.org/our-mission/). Obtaining a high-resolution map of the world's seafloor is
crucial to understanding how oceans work, from geodynamics and geohazards aspects, to the interactions between seafloor
morphology and bottom-current dynamics, and to the distribution and ecological status of benthic habitats. In the last 40 years,
almost two-thirds of marine environments have been "severely altered" by human activity (Díaz et al., 2019) resulting in
significant biodiversity loss and erosion of the ecological services and goods (Worm et al., 2006). In this context, the European
Union has implemented a governance framework specifically aiming at assessing, monitoring, and preserving the status of the
marine benthic natural heritage (Marine Strategy Framework Directive MSFD, 2014/89/EU), but also at promoting the
sustainable exploitation of marine and coastal resources (European MSP Directive, 2008/56/EC). Among the European Seas,
the Mediterranean Sea is a hotspot of biodiversity, hosting more than 7.5% of global biodiversity (Bianchi and Morri, 2000)
with a high percentage of endemic species (Myers et al., 2000) and unique ecosystems. However, the basin is recognized to
be "under siege" due to the historical and still ongoing impacts from multiple stressors such as littering and dumping, trawling,
ghost fishing, seaborne traffic and modification of the seafloor (Coll et al., 2012; Puig et al., 2012; Madricardo et al., 2017,
2019; Canals et al., 2021; Budillon et al., 2022; Pellegrini et al., 2023; Trincardi et al., 2023). This is particularly evident in
the Gulf of Naples, a densely populated coastal region stretching along 385 km on the eastern Tyrrhenian Sea, which represents
an important tourist destination including the Gulf Islands (Capri, Ischia and Procida), Sorrento Peninsula, Vesuvius National
Park, Phlegraean Fields and archaeological sites of Pompeii, Herculaneum, Pozzuoli and Cuma.
The underwater landscape of the Gulf of Naples is geomorphologically complex, with large canyon systems, marine landslides,
debris flow deposits, volcanic apparatuses; the area includes various benthic habitats of ecological relevance from the shore to
the deep sea, such as *Posidonia oceanica* meadows (e.g., MATTM, 2004), animal forests (e.g., Bavestrello et al., 2014), cold-
water corals (CWC, Taviani et al., 2019; Angiolillo et al., 2023), and hydrothermal vent communities (e.g. Apolloni et al.,
2020; Donnarumma et al., 2019). The gulf region also hosts numerous archaeological and cultural heritage sites, threatened
by natural and human pressures (Mattei et al. 2019). To preserve marine biodiversity and the historical value of the area, four
Marine Protected Areas (MPAs) have been established: the Underwater Parks of Baia and Gaiola MPAs, the Regno di Nettuno
MPA and the Punta Campanella MPA (Apolloni et al., 2018).
The first extensive high-resolution mapping of the seafloor of the gulf was performed in the framework of the Italian geological
mapping research program (1997-2017), through bathymetric surveys of the continental shelf/slope system of the Campania





region using numerous multi beam echosounder systems (MBESs) with an average vertical resolution of < 0.25% of the water
depth and position accuracy better than 10 m. The data, acquired at different resolutions, were merged to create a Digital
Terrain Model (DTM) with a homogeneous grid and with a cell spacing of 20 m (Aiello et al., 2020). This map highlighted
the most prominent geomorphological features in the coastal zone such as the canyons, banks, debris avalanches, hydrothermal
vents and volcanoclastic basement outcrops with high ecological value habitats in urgent need of preservation (Taviani et al
2019). This valuable dataset was shared in gridded form, within the EMODnet project, as 1/16 arc minutes (ca. 115 m) DTMs.
High-resolution data for selected areas are also available as 1/128 or 1/256 arc minutes (ca.15 m or 7 m) HR-DTMs
(https://emodnet.ec.europa.eu/geoviewer/).
Despite the significant effort of ongoing national and international projects and infrastructures worldwide to make data
available, such as GEBCO (https://www.gebco.net) and EMODnet (https://emodnet.ec.europa.eu/en), local high-resolution
datasets and raw data are typically not yet accessible (Sievers et al., 2021). Indeed, databases are often generated, hosted, and
administered by various institutes in the world with dissimilar data policies, which often do not follow the Findable, Accessible,
Interoperable and Reusable (FAIR) data principles (Stall et al., 2019).
This study presents the results of a high-resolution geophysical survey conducted in October 2022 on board R/V Gaia Blu
using three different state-of-the-art MBESs (Kongsberg EM 2040, EM 712, and EM 304) and aims at improving the
knowledge of the seascape of the Gulf of Naples by enhancing the analysis/visualization of seabed morphology through high-
resolution digital bathymetric models.
Given the unprecedented high- and multi-resolution survey conducted in the study area and the availability of ancillary data
such as backscatter and water-column data, this dataset represents a unique benchmark for future studies related to geohazards
assessment, sediment transport, fishery management, resource exploration and sustainable exploitation, maritime spatial
planning and decision making, marine ecosystem and habitat mapping, oceanographic modeling including storm surges and
scenarios of tsunami wave propagation.
We discuss the quality of the data collection (section Data Quality) and present three examples that highlight the potential
applications of this dataset (Section Results and discussion). Our contribution also aims at highlighting the innovative approach
used during JammeGaia22 (Section Multibeam data processing), where data are processed daily on board and can be made
available to the scientific community and the generic public in near real-time via a geoportal, making the datasets FAIR and
facilitating interdisciplinary research within the Open Science Principles.

**2 Study area - Geological and geomorphological background**

The investigated area belongs to the central-eastern margin of the Tyrrhenian Sea, encompassing the region between the



western margin of the Southern Apennines thrust belt and the Tyrrhenian abyssal plain (ca.3000 m deep; Figure 1). The
Tyrrhenian Sea is the youngest back-arc basin of the Mediterranean Sea that developed since the Middle Miocene (Trincardi
and Zitellini, 1987; Kastens et al., 1988; Lymer et al., 2018; Loreto et al., 2021; Miramontes et al., 2023) reflecting the east-
and south-eastward retreat of the Ionian slab, guided by the Africa-Europe convergence (Moussat et al., 1985; Malinverno and
Ryan, 1986; Kastens et al., 1988). The Campania segment of the eastern Tyrrhenian margin is characterized by a series of NE-
SW trending half-graben bounded by structural highs that have developed since the early Pleistocene and accommodate the
tectonic-controlled subsidence of the alluvial plains along with their submerged counterparts, namely the Gaeta Gulf, the Gulf
of Naples and the Gulf of Salerno (Figure 1; Romano et al., 1984; Ruberti et al., 2022; Amato et al., 2011; Bellucci et al.,
103   2006).

Structural lineaments also control the preferential pathways of volcanic activity, particularly in the last 2 My. Volcanic activity
followed an eastward migration, governing the geomorphological setting of the region and promoting deposition of
sedimentary sequences up to 3 km thick (Milia, 1999; Milia et al., 2003).The Phlegraean Fields volcanic area is a 78-ka old
active poly-calderic system (Scarpati et al., 2012) that has affected its territory in the last millennia and has strongly influenced
the evolution of the adjacent coasts during the late Pleistocene and Holocene, which has been mainly shaped by three super-
eruptions. The oldest one was the Campanian Ignimbrite (CI) eruption that occurred at ca. 35-40 ka BP (Giaccio et al., 2017).
After this main event, the northern part of the just-formed caldera was submerged by the sea. The second eruption, which led
to the formation of the Masseria Del Monte Tuff, occurred at 29.3 ka BP (Albert et al., 2019). The Neapolitan Yellow Tuff
(NYT; Deino et al. 2004) eruption at ca. 15 ka BP contributed to the formation of the youngest caldera (Orsi et al., 1992),
nowadays well documented also offshore (Sacchi et al., 2014; Steinmann et al., 2016, 2018). Besides volcanic eruptions,
alternating long-term magma/hydrothermal fluid inflation and deflation processes controlled the morphological evolution of
this area. Further, short-term vertical, meter-scale, ground movements characterised times immediately preceding and
following each eruption, which produced rapid relative sea-level variations along the entire coastal sector (Isaia et al., 2019
and reference therein). The area has experienced high rates of subsidence (approx. 4.0 mm/yr) through the Pleistocene
(Torrente et al., 2010; Milia et al., 2017; Iannace et al., 2018), accompanied by the activity of major NE–SW-striking faults.
At present, intense seismicity, including the Md 4.0 earthquake occurred on 2nd October 2023, is instead associated to the
18.0 mm/yr uplift of the central portion of the Phlegraean Field area.
Volcanic activity, long-term vertical ground movements, glacio-eustasy and the rapid dismantling of the emerging landscapes
have driven a rapid geomorphological evolution of the margin, resulting in steep slopes, canyoning, deep-sea fan accretion and
gravitational slope instability. Extensive lateral collapses of the volcanic edifices have been documented offshore, south of
Ischia Island (Chiocci et al., 1998; Chiocci and de Alteriis, 2006; de Alteriis et al., 2010), possibly occurred also in historical
time, and two others of minor extent to the west and north of Ischia Island (Budillon et al., 2003; Violante et al., 2003) and in
the Gulf of Naples (Milia et al., 2008, 2012; Passaro et al., 2018). The rapid aggradation of volcaniclastic deposits in shallow
marine environment and the entrance of pyroclastic flows into the seawater also led to seafloor instability and creep in the
prodelta offshore the main rivers (Sacchi et al., 2005; 2009).
Three main turbiditic systems, namely Cuma, Magnaghi and Dohrn Canyons, and the deep structurally controlled Salerno
Valley, have developed along with the rising of intra-slope reliefs and volcanic activity, and acted as main conduits delivering
sediment towards deeper-water domains (Passaro et al., 2016). These features characterize the present-day seafloor
morphology and, although partially inactive, are of paramount interest as hotspots of biodiversity in the Mediterranean Sea
(e.g., Taviani et al., 2019; Mussi et al., 2022).




Figure 1. Map of the study area in the central Tyrrhenian Sea showing the main physiographic and tectonic features (modified from Aiello

136          et al., 2020). Elevation and bathymetry from EMODnet bathymetry (https://emodnet.ec.europa.eu/en/bathymetry).




## 3. Materials and methods

### 3.1 Multi beam data acquisition

Multi beam data were collected during the JammeGaia22 cruise from September 27[th] to October 20[th] 2022 using three different MBES: the Kongsberg EM2040-04 MKII 0.4°x0.7° suited for water depths between 50 and 150 m, Kongsberg EM712 1°x0.5° for water depths between 150 and 1000 m and Kongsberg EM304 MKII 1°x1° for water depth greater than 1000 m (Table 1for acquisition settings).

Table 1. Acquisition settings for the three multi beam echosounder systems.

| MBES | Water depth (m) | Frequency (kHz) | Angular coverage (degree) | Ping rate (Hz) | Acquisition mode |
|---|---|---|---|---|---|
| EM2040 | 50-100 | 300 | 65 | 1.5 | Deep |
| EM2040 | 100-150 | 200 | 70 | 1.5 | Very deep |
| EM712 | 150-600 | 70-100 | 70 | 2 | Shallow |
| EM712 | 600-1000 | 40-100 | 70 | 2 | Deep |
| EM304 | >1000 | 30 | 65 | >5 | Auto |

The MBESs were hull-mounted on the R/V GAIA BLU gondola with a T-configuration of linear transducer arrays. A Seapath 380 system was used for ship positioning, supplied by a Fugro HP differential Global Positioning System (DGPS), with Marinestar GNSS signal accuracy better than 5 cm. The Kongsberg motion sensor MRU (Motion Reference Unit) 5 and a Dual Antenna GPS integrated into the Seapath, were used to correct for pitch, roll, heave and yaw movements (reaching 0.02° roll and pitch accuracy, and 0.075° heading accuracy). A Valeport mini SVS sensor was positioned close to the transducers to continuously measure the sound velocity for the beamforming. Sound velocity profiles (SVPs) were systematically collected at least twice a day with a Valeport Midas SVP, for a total of 40 SVPs. Data were logged, displayed and checked in real-time by the Kongsberg data acquisition and control software SIS 5 (Seafloor Information System). A software tool was used to extend the SVPs down to 1200 m water depth. Since the Mediterranean Sea is characterized by a stratified water column with peculiar changes in the physical-chemical properties (Tanhua et al. 2013; Rossi et al. 2014; Basterretxea et al. 2018), a linear regression based on the collected SVP data was run in R software (R Core Team, 2019) to estimate the sound velocity values down to 12000 m depth.

Professional topographers measured the offsets of the instruments with millimetric accuracy using a dedicated dimensional survey of the ship's hull at dry dock.

Sensors have been calibrated during the Sea Acceptance Tests (roll, pitch, time and heading offsets) and were also regularly checked in post-processing (Table 2 for calibration values).



Table 2. Calibration values applied after the Sea Acceptance Test.

| MBES | Pitch | Roll | Heading |
|---|---|---|---|
| EM2040 | +0.10° | +0.5° | -0.20° |
| EM304 | 00.00° | +0.2° | 0.00° |
| EM712 | -0.10° | -0.07° | -0.15° |


We kept a 20% overlap between lines, avoiding the influence of external beams of bad quality given by possible residual errors
in roll and sound speed profile measurements. The multi beam operated with an average swath opening angle of about 65°/70°
(Table 1) for each multi beam system. The vessel sailed with a reasonably constant speed of 8 knots, considered ideal to have
the minimum noise and tested during the Sea Acceptance Test. Seafloor and water column backscatter data were collected
simultaneously during bathymetric data acquisition.

**3.2 Multibeam data processing**
The bathymetric data collected were immediately processed on-board to produce DTMs and backscatter mosaics, uploaded
daily in a dedicated WebGIS to inform the scientific community on the progress of the campaign and make the data openly
available. The data processing workflow is summarized in Figure 2.

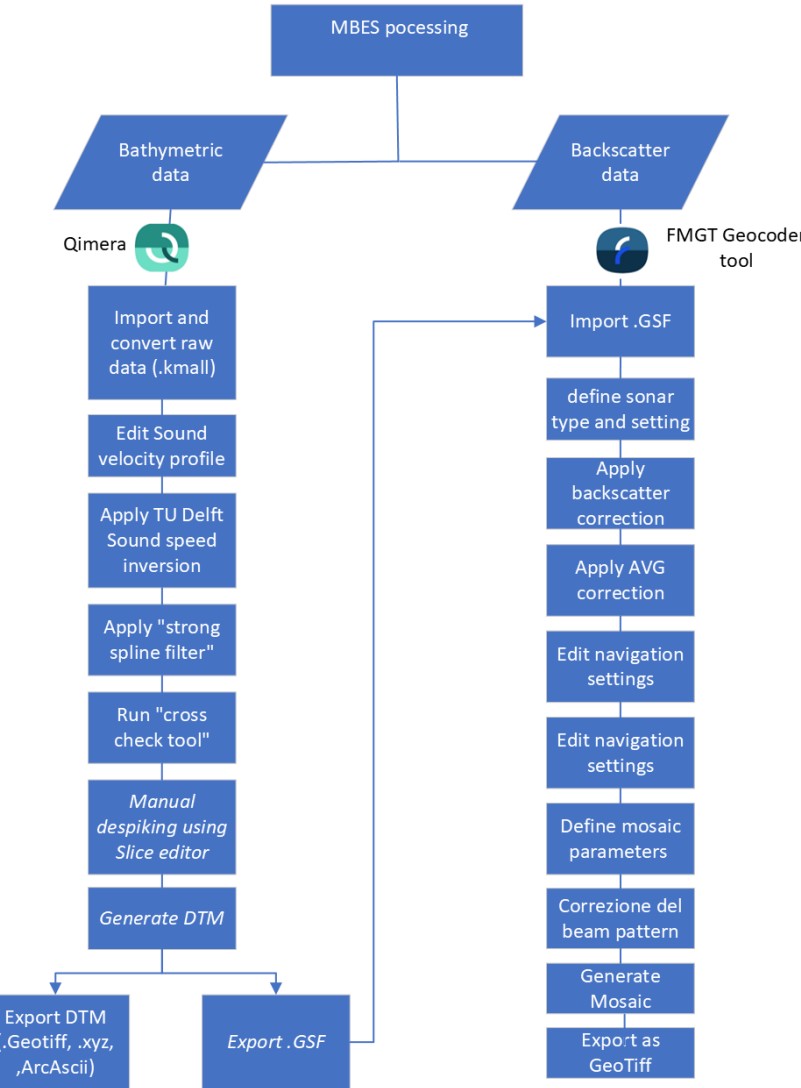

Figure 2. Workflow of bathymetric and backscatter data processing.

### 3.2.1 Bathymetric data processing

The processing of the raw data was carried out using the QPS Qimera v.2.5.0 software (Quality Positioning Services BV, Zeist, Netherlands) following a standard procedure, which includes sound speed correction, removal of erroneous soundings, and correction of vertical offsets. The quality of the data was initially checked using the 'Cross Check Tool' to check for soundings with significant offsets from the local mean water depth.



When sound velocity errors were evident in the data, the TU Delft Sound Speed Inversion tool (Beaudoin et al., 2018) was
used to correct the profile. The tool applies an algorithm that allows a completely automated refraction error correction. It
works by taking advantage of the overlap between survey lines to simultaneously estimate sound speed correction for a given
set of pings and their neighbours, by computing a best-fit solution that minimizes the mismatch in the areas of overlap between
lines (Mohammadloo et al., 2019). The settings applied for TU Delft Sound Inversion were data-specific, depending on the
quality of the SVP, upon initial assessment. Nevertheless, we typically applied the 'Quick Search' algorithm and 'Coarse'
configuration as an initial setting, and we then applied adjustments if necessary.
After the sound speed correction, the strong spline filter of Qimera allowed removal of soundings beyond the local mean water
depth (offsets); the remaining offsets (if any) were removed manually using the 'Slice editor' of Qimera. The processed
bathymetric data were exported into GSF format for backscatter processing and to a gridded surface data (GeoTIFF). The
resolution of the GeoTIFF was defined based on the water depth and the footprint calculated for each sonar used (Table 3).

Table 3. Calculated footprints of ensonified seafloor area at different water depths for each MBES. Products and dataset are available at
section Data Availability.

| MBES | Water Depth (m) | TX Footprint (m) | RX Footprint (m) | Insonifed area (mq) | Grid resolution (m) |
|---|---|---|---|---|---|
| **EM2040 (0.4°x 0.7°)** | 50 | 0.4363 | 0.6109 | 0.92 | **2** |
| | 60 | 0.5236 | 0.7330 | 1.10 | |
| | 70 | 0.6109 | 0.8552 | 1.28 | |
| | 80 | 0.6981 | 0.9774 | 1.46 | |
| | 90 | 0.7854 | 1.0996 | 1.65 | |
| | 100 | 0.8727 | 1.2217 | 1.83 | |
| **EM712 (0.5°X1°)** | 150 | 1.3090 | 2.6181 | 3.28 | **5** |
| | 200 | 1.7453 | 3.4907 | 4.37 | |
| | 300 | 2.6180 | 5.2361 | 6.56 | **10** |
| | 400 | 3.4907 | 6.9815 | 8.75 | |
| | 500 | 4.3634 | 8.7269 | 10.94 | |
| | 600 | 5.2360 | 10.4722 | 13.12 | **15** |
| | 700 | 6.1087 | 12.2176 | 15.31 | |
| | 800 | 6.9814 | 13.9630 | 17.50 | **20** |
| | 900 | 7.8540 | 15.7084 | 19.69 | |
| | 1000 | 8.7267 | 17.4537 | 21.87 | |





| | 1000 | 17.4537 | 17.4537 | 30.94 | **30** |
| | 1100 | 19.1991 | 19.1991 | 34.03 | |
| | 1200 | 20.9445 | 20.9445 | 37.12 | |
| **EM304 (1°X1°)** | 1300 | 22.6899 | 22.6899 | 40.22 | **40** |
| | 1400 | 24.4352 | 24.4352 | 43.31 | |
| | 1500 | 26.1806 | 26.1806 | 46.40 | |
| | 1600 | 27.9260 | 27.9260 | 49.50 | |
| | 1700 | 29.6714 | 29.6714 | 52.59 | |
| | 1800 | 31.4167 | 31.4167 | 55.68 | |
| | 1900 | 33.1621 | 33.1621 | 58.78 | |
| | 2000 | 34.9075 | 34.9075 | 61.87 | |

### 3.2.2 Backscatter data post-processing

The MBES backscatter data were processed using the QPS Fledermaus Geocoder Tool (FMGT) v.7.10.2 software. The processed MBES data (.gsf) were used to apply backscatter corrections, beam pattern correction, and angle-varying gain (AVG) corrections to the backscatter data. After these corrections, FMGT applied the sonar's navigation data (i.e., XY coordinates, roll, heading, pitch, heave) to improve the spatial accuracy of the data. The DTM generated in Qimera provided a reference grid to improve backscatter corrections. The reference grid was included by the FMGT software to determine topographic slope, while the corrected bathymetry in the source files (i.e., GSF) was regularly used to geo-reference the snippet trace from a single ping to the correct spot on the seafloor and with the correct scaling (Quality Positioning Services B.V., 2020). Finally, the backscatter snippets were mosaicked with the 'No Nadir possible, 25% overlap' algorithm to reduce the banding effect, and 30-40% line blending was applied to blend the pixels in the overlapping areas. The mosaics were gridded in various resolutions (Table 4) with dB values cropped to $\pm\,3\sigma$ and logarithmically mapped to 8-bit scale. These mosaics were exported as 'One merged Colored GeoTIFF format'.

Table 4. Resolution of backscatter mosaic for each MBES. Products and dataset are available at section Data Availability.

| MBES | Mosaic resolution (m) |
|---|---|
| EM2040 | 5 m |
| EM712 | 10 m |
| EM304 | 30 m |






### 3.3 Bathymetric derivatives


A geomorphometric analysis of the seabed was carried out using ArcGIS to emphasize any subtle variation in seafloor
morphology. The geomorphometric indices calculated were slope, broad-scale and fine-scale Bathymetric Position Index
(BPI), and rugosity.
The slope is a first-order derivative of the bathymetry and represents seabed maximum inclination (in any direction) in degrees,
using the AreaSlope algorithm, included in Marine Toolbox for ESRI ArcGIS developed by Dr Tim Le Bas (Le Bas 2016),
picking an area of 3x3 pixels around each cell. Values are real numbers between 0.0° and 90.0°, areas of no data have a
conventional value of -1.0. Depth values in input were smoothed before calculation of the slope using a user-defined smoothing
window of 3x3. This approach served to removed local changes giving a regional value for slope and diminishing edge effect
(Dolan, 2012).
Broad- and fine-scale BPIs were calculated using Benthic Terrain Modeler (BTM) toolbox for ArcGIS (Walbridge et al., 2018;
Lundblad et al., 2006). BPI is a second-order derivative (as it is derived from the slope) of the bathymetry and is modified
from topographic position index as defined by Weiss (2001) and Iampietro and Kvitek (2002). It evaluates differences in
elevation between a focal point and the mean elevation of the surrounding cells within a user-defined window. Values range
from -1 to +1, with negative values reflecting depressions in the seabed, null values for planar areas and positive values
denoting the reliefs. Broad-scale BPI allows the identification of main regional features within the seafloor, while fine-scale
BPI helps identify smaller features of the benthic landscape. The values used to calculate BPIs for all the bathymetric surfaces
are reported in Table 5.

### 4. Results and Discussion


### 4.1 Multi-resolution bathymetric grid


The multi-resolution grid covers an area of about 5000 km$^2$ offshore the Gulf of Naples from 50 to more than 2000 m water
depth (Figure 3). The different resolutions, depending on the water depth and the MBES footprint, of the acquired data reveal
the complexity of the seafloor with unprecedented details and allow to better discriminate geomorphological features already
described in the literature (D'Argenio et al., 2004).


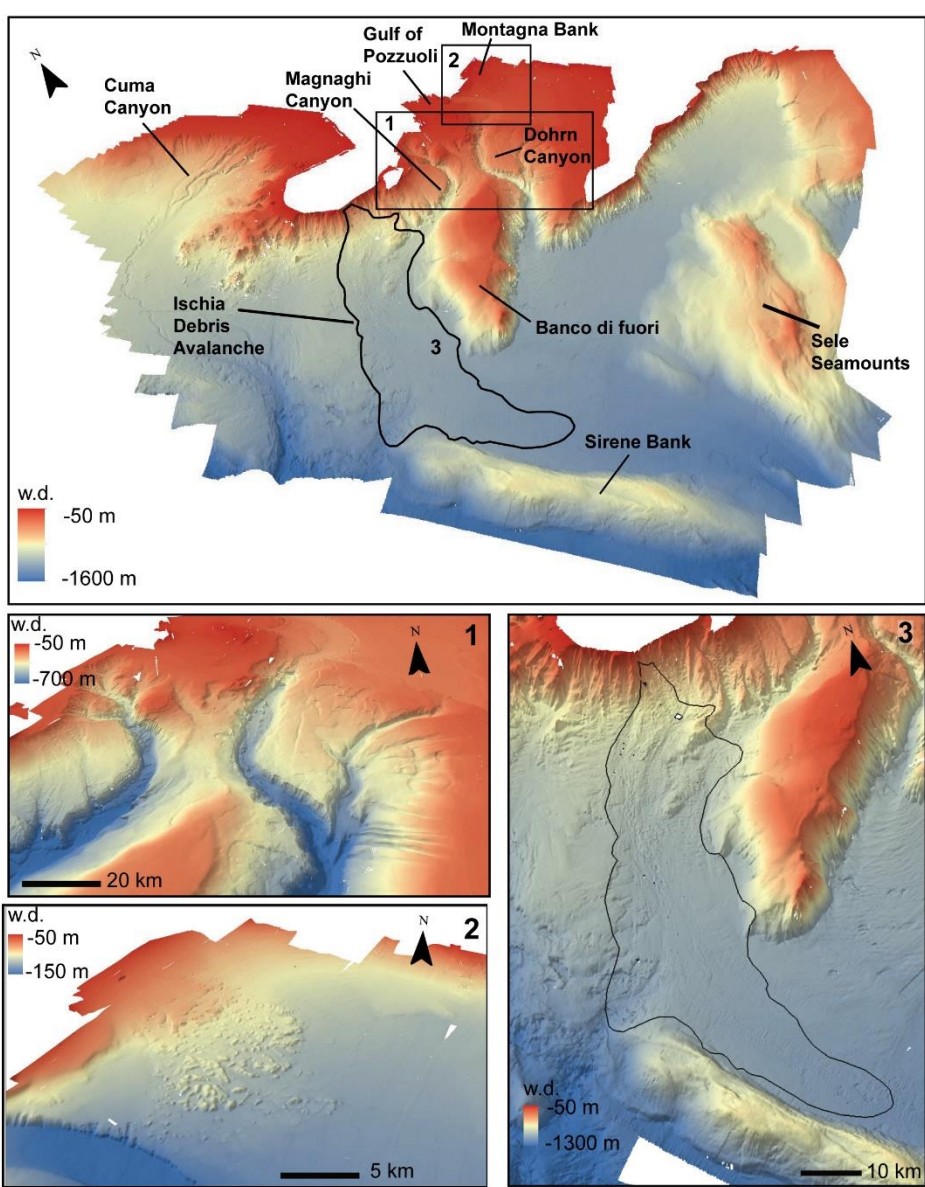


Figure 3. Bathymetric map of the study area (20 m resolution, 2 vertical exaggeration) showing the main seabed features; (1) multibeam bathymetry (20 m resolution, 2 vertical exaggeration) of the Dohrn and Magnaghi canyon systems; (2) multibeam bathymetry of the Montagna Bank area; and (3) multibeam bathymetry of the debris avalanche offshore the Ischia Island.

Coupled with other indices, this high-resolution bathymetry not only is valuable information to study sediment dynamics, and morphotectonics of canyons, structural highs and seamounts, but also represents a baseline to investigate the presence and distribution of benthic habitats and infer hydrological transients hugging the sea floor. To demonstrate how the newly acquired



data allow to appreciate the variations of the seafloor, broad- and fine-scale BPI were calculated from the bathymetry in three
selected sectors of the study area using the parameters reported in Table 5.

250       Table 5. Inner and outer radius used for calculation of Bathymetric Position Index (BPI) for selected areas by depth range.

| Area | Depth range (m) | Resolution (m) | Broad-scale BPI Inner – outer radius (cells) | Fine-scale BPI Inner – outer radius (cells) |
|---|---|---|---|---|
| Canyons of the Gulf of Naples | 50-100 | 2 | 30-60 | 2-5 |
| | 101-200 | 5 | 12-30 | 2-5 |
| | 201-500 | 10 | 6-15 | 2-5 |
| | 501-700 | 15 | 4-9 | 2-5 |
| | 701-1000 | 20 | 3-8 | 2-5 |
| | 1001-2500 | 30 | 2-5 | 2-5 |
| Montagna Bank | 50-100 | 2 | 30-60 | 5-8 |
| | 101-200 | 5 | 12-30 | 5-8 |
| | 201-500 | 10 | 6-15 | 5-8 |
| Ischia debris avalanche | 50-100 | 2 | 30-60 | 1-3 |
| | 101-200 | 5 | 12-30 | 1-3 |
| | 201-500 | 10 | 6-15 | 1-3 |
| | 501-700 | 15 | 4-9 | 1-3 |
| | 701-1000 | 20 | 3-8 | 1-3 |
| | 1001-1900 | 30 | 2-5 | 1-3 |


## 252 4.1.1 Canyons of the Gulf of Naples

The morphology of the Dohrn and Magnaghi Canyons is possibly controlled by the presence of extensional faults coupled
with the volcanic activity characterising the area. Both canyons acted as large drainage systems within this proximal marine
area during the Late Quaternary (Aiello et al., 2020 and references therein). The two branches of Dohrn Canyon are about 500
m wide and show a V-shaped profile in the upper part and a U-shaped profile in the lower part, suggesting uniform sediment
fill of the thalweg. The bathymetric derivatives confirm the complexity of these drainage patterns showing some differences
possibly related to the stratigraphy of the eroded terrains and to the recurrence and or competence of the flows flushing the
two systems: straight gullies characterise the flanks of Dohrn Canyon and normally do not indent the outer shelf, with the
exception of the area NW of Capri (Fig. 4). Canyon Dohrn emanates from Ammontatura channel, on the inner shelf, a possibly
active sediment conduit also during sea level rise and high stand conditions; Dohrn Canyon undercuts its secondary branch


located north of Capri Island under-excavating its base by 50m. The straight gullies on the flanks of Dohrn Canyon are hanging
above the canyon thalweg suggesting the activity of powerful flows along the axis of the canyon. Moreover, the fine-scale BPI
highlights terrace rims along Dohrn Canyon flanks and slide scars with a slide deposit at their foot (Aiello et al., 2020), as well
as the gullies with head scarps and along-slope small-scale sand splays located on the southern flank of Banco di Fuori. Dohrn
Canyon shows a radial bedform field in its lower portion where the canyon broadens, and its floor decreases its gradient.
Comparison with pre-existing data in this area suggests that the bedform field has not moved in the last two decades.
In contrast, Magnaghi Canyon is shorter, less deeply incised and not gullied on its flanks, possibly reflecting its lack of
connection to a major source of sediment-laden flows. The right-hand side of the canyon shows short and straight incisions
with marked bedforms hat appear reminiscent of cyclic steps (Kostic, 2011; Slootman and Cartigny, 2020) and can be clearly
discerned on the slope map and on the DPI maps.

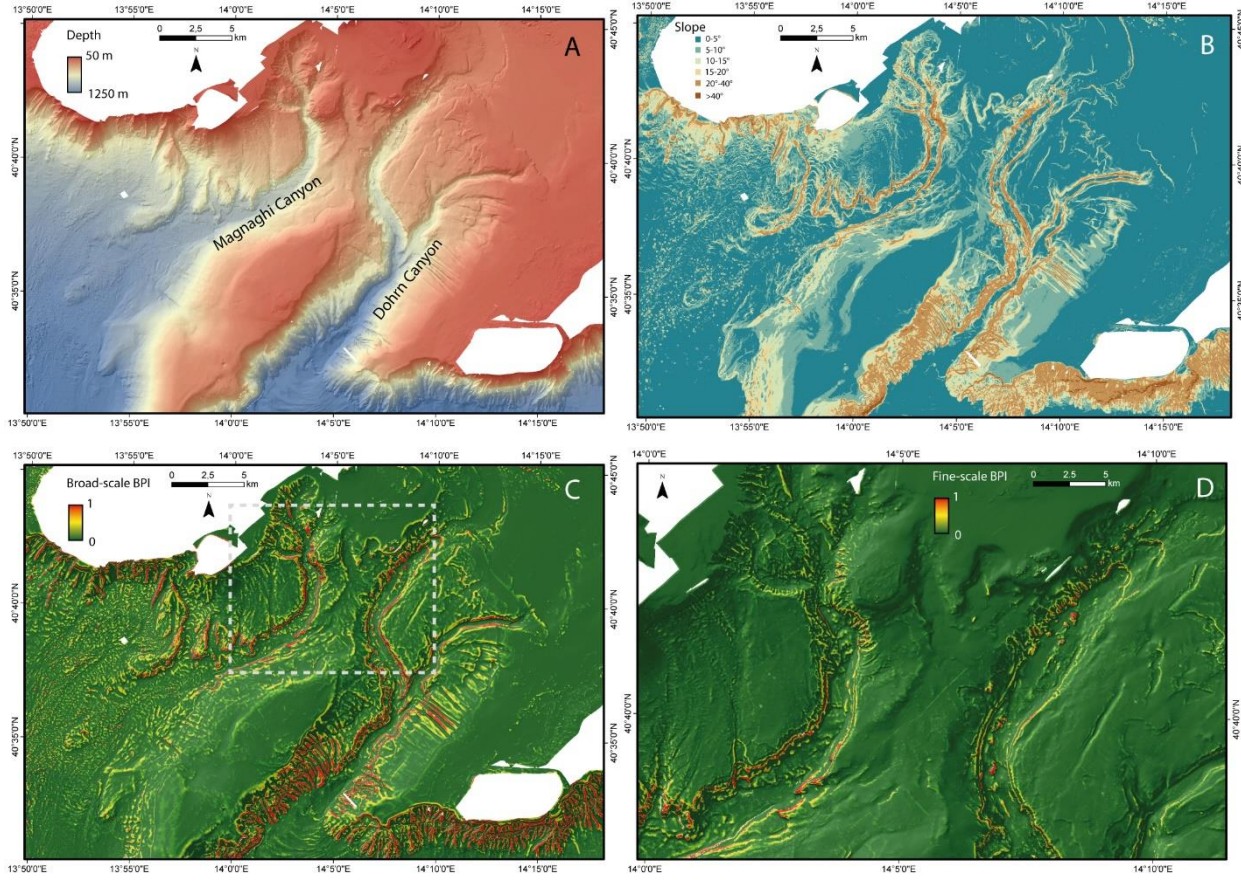


Figure 4 (A) Bathymetric data of canyons of the Gulf of Naples; (B) Slope; (C) positive values of broad-scale and (D) fine-scale BPI of a
portion of the area (dashed rectangle in C) calculated from the newly acquired multi-resolution grid, showing the drainage pattern of the
Dohrn and Magnaghi canyons.





### 4.1.2 Cuma Channel

Cuma Channel is a complex sediment conduit characterized by 1) an upper section, between the shelf-edge and the base of Gaeta basin, where three independent sub parallel channels present gullied heads, low sinuosity and flat channel floor; 2) a relatively narrow thalweg characterize by a prominent high sinuosity on the sub-horizontal floor of Gaeta Basin and 3) a straighter channel, proceeding in deeper waters across the steepening slope region.

Peering both bathymetric and backscatter images prompt several questions that will be worth addressing in future cruises, after collecting complementary core and seismic-stratigraphy data. In particular:

1) there is no continuity between either of the three channels dissecting the upper slope and the high sinuosity channel on the floor of Gaeta basin; however, backscatter images hint to a seaward continuity of the most meridional of the three slope channels characterized by higher backscatter and, likely, coarser grained sediment. This channel reaches a north-south orientation before widening and rapidly reducing its seafloor reflectivity;

2) the high sinuosity to the west is therefore disconnected from its original feeder, upslope, and, proceeding downslope, bends gently to the Southeast and then to the Southwest in the lowermost tip of the mapped area; interestingly, the region located west of this gentle, multi-kilometric, bend is carved by several barchan-like scours that can be hypothetically ascribed to overflows of a much larger volume compared to the size of the channel conduit;

3) knowing that the Volturno prodelta has reached the shelf edge, it is possible that hyperpycnal flows from the river ignite flows on the slope that are capable to hug the seafloor and reshape its morphology, as documented during the modern sea level high stand in some other example of high discharge systems like the Crati River (Lucchi et al., 1983).

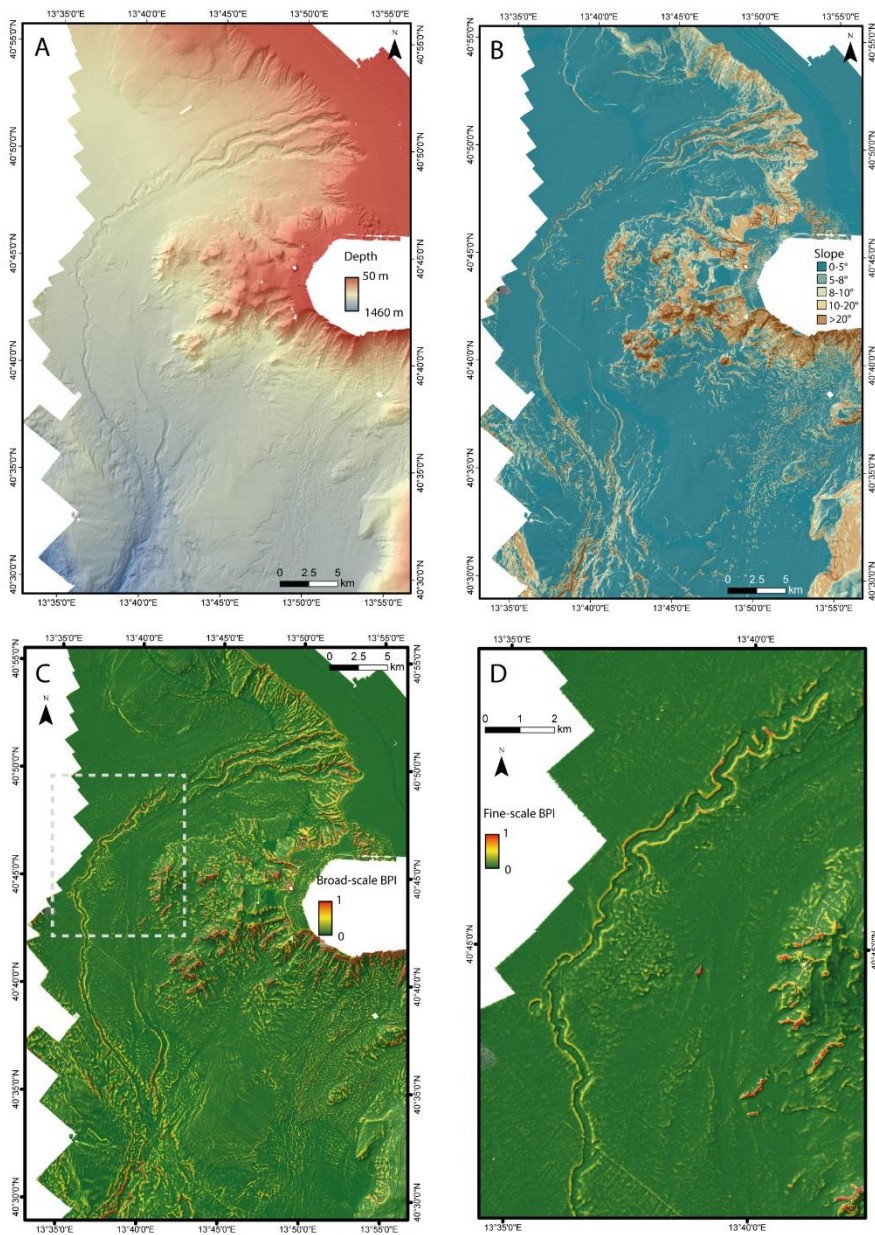

Figure 5 (A) Bathymetric data of the Cuma channel; (B) Slope; (C) positive values of broad-scale and (D) fine-scale BPI of a portion of the area (dashed rectangle in C) calculated from the newly acquired multi-resolution grid.

### 4.1.3 Montagna Bank

In the shallower area of the Gulf of Naples, **Montagna Bank** is a morphological high extending over 25 km$^2$ (Passaro et al. 2014, 2016, 2018; Ventura et al. 2016), where volcanoclastic materials (dominantly low-density pumice) underwent small-scale deformation leading to the growth of meter-scale sediment-diapirs and possible fluid-escape features; in particular, this

hummocky area includes 280 mounds, 650 cones with meter-scale hight, and 30 pockmarks (Sacchi et al., 2019), between
100 and 150 m water depth. The slope calculated for the Montagna Bank shows the inclinations of both the whole
morphological high and of the individual bedforms surrounding it (i.e., the flanks of the Ammontatura channel and sedimentary
bedforms located W of the Montagna Bank). Furthermore, the calculated BPIs reveal large and small mounds constituting the
hummocky-like morphology of the large-scale relief.

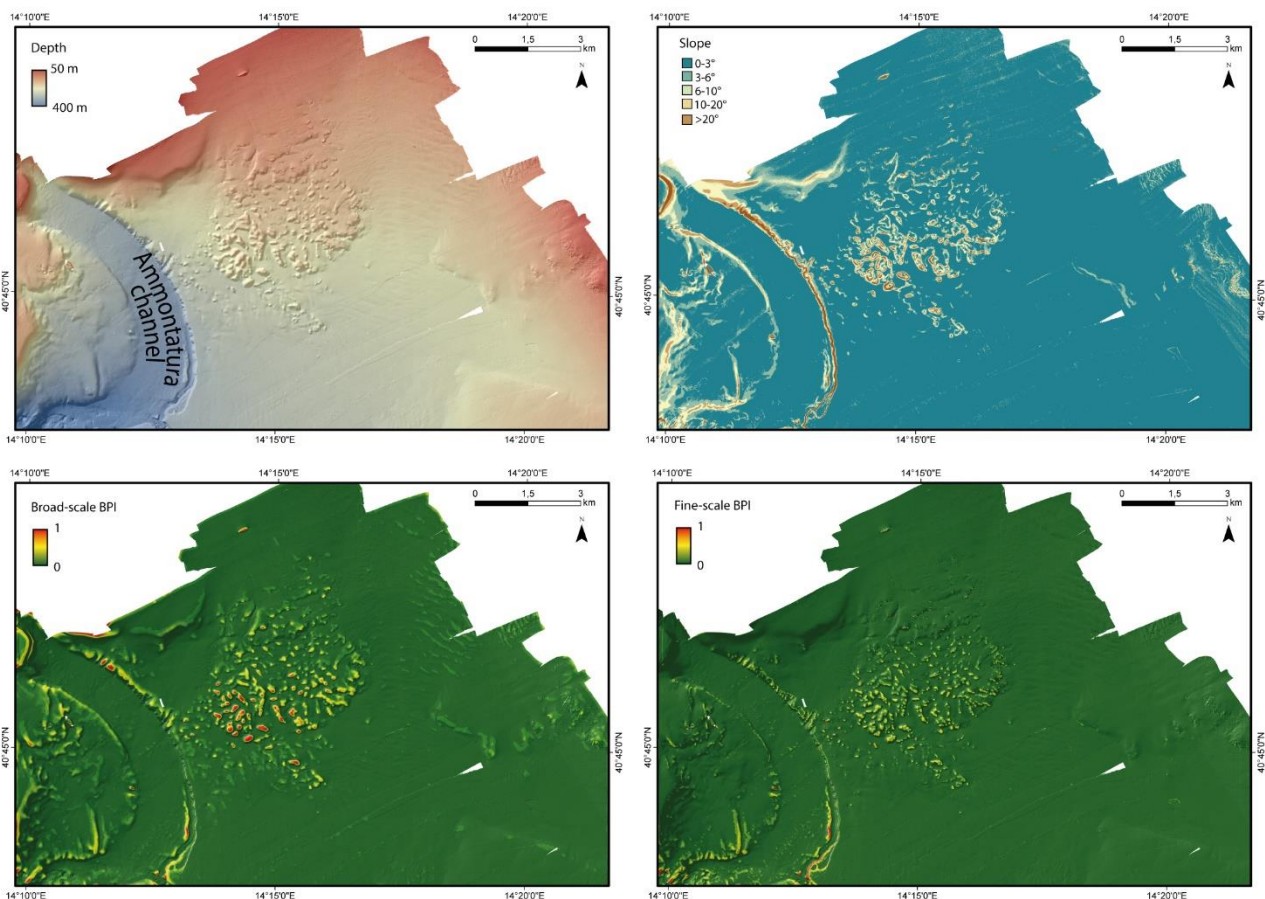

Figure 6. (A) Bathymetric data of the Montagna Bank; (B) Slope; (C) broad-scale and (D) fine-scale BPI calculated from the newly
acquired multi-resolution grid, showing the morphology of the Montagna Bank.


### 4.1.4 Ischia debris avalanche

The Ischia debris-avalanche is located south of Ischia Island and is a 50-km-long tongue characterised by a hummocky
topography extending for about 200 km$^2$ with fields of giant blocks spanning in size from a few metres to > 200 m across and
with larger blocks being up to 30–50 m high (Chiocci and de Alteriis, 2006; de Alteriis et al., 2010). The hummocky deposit



follows the local pre-collapse topography, and, on its eastern side, it overflows into the Magnaghi Canyon. The slope (Fig. 6B), the broad-scale (Fig. 6C), and fine-scale (Fig. 6D) BPI obtained using different inner and outer rays (Tab. 5), calculated from the newly acquired bathymetric data, allow to better appreciate the morphology of the deposits and clearly identify individual debris blocks, allowing better measurement of their size and volume.

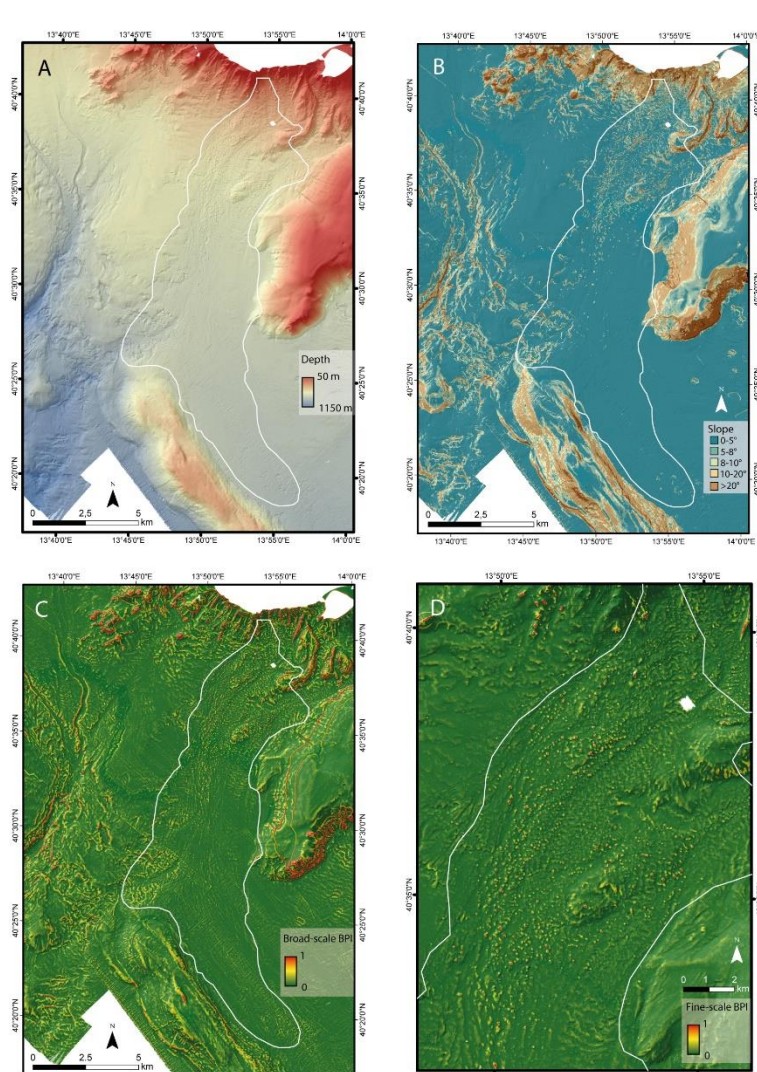

Figure 7. (A) Bathymetric data of the Ischia Debris Avalanche; (B) Slope; (C) broad-scale and (D) fine-scale BPI calculated from the newly acquired multi-resolution grid, showing the location and morphology of debris blocks. The white square delimitates the area that contains the debris avalanche.



## 4.2 The multi-resolution backscatter mosaic

The backscatter intensity data acquired during the JammeGaia22 cruise represent the first dataset covering the entire Gulf of Naples, Ischia surroundings, Salerno Valley and Sirene Smt. Three mosaics were exported at different spatial resolutions: 5 m for the dataset acquired using the EM2040 system, 10 m for EM712 and 30 m for EM304 (Figure ).

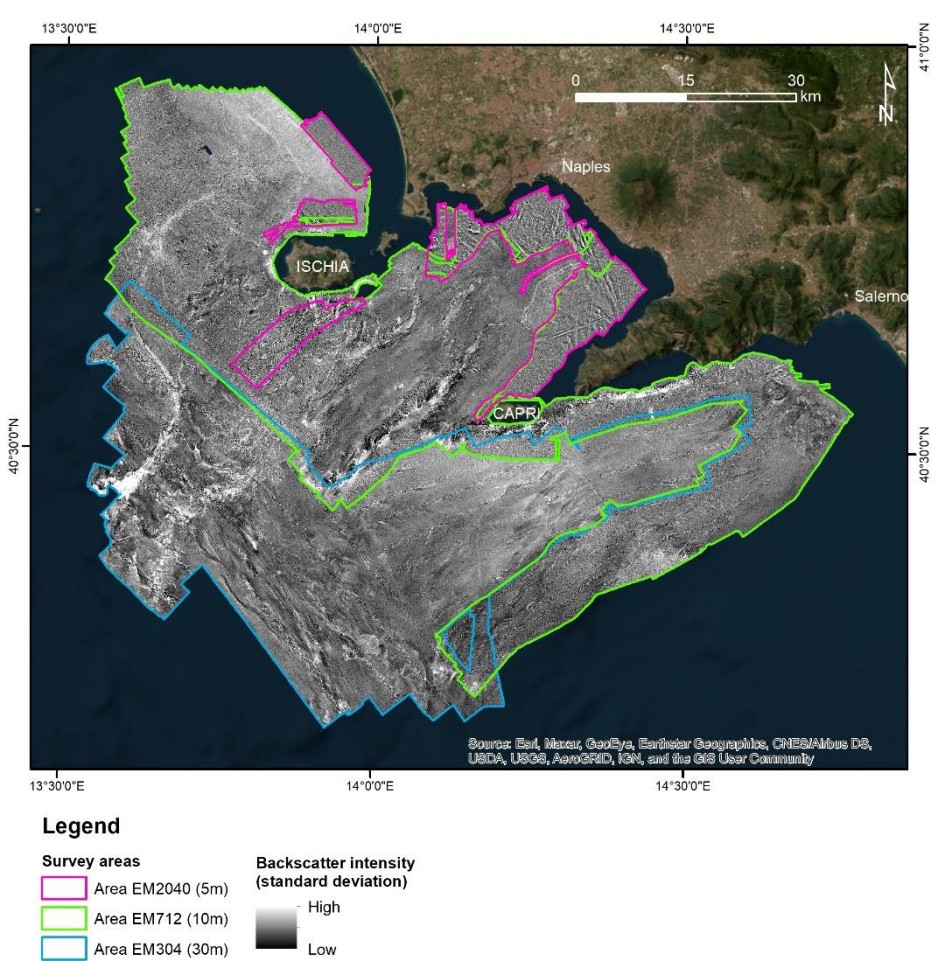

Figure 8. Backscatter mosaics acquired during the JammeGaia22 cruise with the survey areas covered by the three MBES.

Details are shown for four areas: Montagna Bank, Sorrento peninsula, north and west Ischia Island, and Magnaghi canyon head (Figure 8). The backscatter highlighted the hummocky-like morphology of the Montagna Bank and the trawl marks on the seabed around it. The backscatter dataset of the Sorrento peninsula revealed the occurrence of patterns likely associated with coralligenous bioconstructions (the lighter areas) and seagrass meadows along the coast, as previously highlighted in other studies (CARG - Geological CARtography project; EMODnet Seagrass cover (Essential Ocean Variable) in European waters (2023); Russo et al. 2008; Buonocore et al., 2020). Also, the hummocky morphology of the debris avalanches occurring

north and west of Ischia Island is enhanced by the seabed reflectivity, together with features of fluid escapes (white spots in
Figure 8C) around Ischia Island and in the head of the Magnaghi canyon, due to the hydrothermal activity characterizing the
area.



Figure 9. Details of the seabed backscatter in different locations: A) Montagna Bank hummocky morphology and  trawl marks (EM2040 –
348        5m); B) Coralligenous bioconstructions west of the Sorrento peninsula (EM712 – 10m); C) debris avalanches north and west of Ischia
Island and fluid escape features (EM712 – 10m); D) head of the Magnaghi Canyon characterized by fluid escape features (EM712 – 10m).




## 4.3 MBES data quality

The uncertainty of the bathymetric data was calculated in Qimera v.2.5.4 according to the IHO Standards for Hydrographic
Surveys 2-44 6ᵗʰ Edition, 2022. Total Horizontal Uncertainty (THU) and Total Vertical Uncertainty (TVU) were calculated
considering the standard deviation offsets of the MRU, MBES, sound velocity probe, and positioning system. Parameters used
for the calculation of THU and TVU were taken from the datasheet of the MBES systems and installation report (Table 6).
The uncertainty values of EM2040 vary depending on the sampling frequency and depth changes during the survey. Hence,
the values presented below are the range of uncertainty calculated for 200 kHz and 300 kHz and different pulse lengths that
were used during acquisition.

360         Table 6. Parameters used to calculate Total Horizontal Uncertainty and Total Vertical Uncertainty

|  | EM2040 | EM712 | EM304 |
|---|---|---|---|
| **Echosounder** |  |  |  |
| Pulse Length | 2, 3, 6, and 12 ms | 2 ms | 7.5 ms |
| Sampling Frequency | 200kHz, 300 kHz, | 70 kHz | 25 kHz |
| **Sound Velocity** |  |  |  |
| SD Surface sound speed | 0.02 m/s | 0.02 m/s | 0.02 m/s |
| **Beam Width** |  |  |  |
| Beam Width Along (Tx) | 0.4° | 0.5° | 1.0° |
| Beam Width Across (Rx) | 0.7° | 1.0° | 1.0° |
| **Offsets (Argo)** |  |  |  |
| SD Roll Offset | 0.04° | 0.04° | 0.04° |
| SD Pitch Offset | 0.02° | 0.02° | 0.02° |
| SD Heading Offset | 0.02° | 0.02° | 0.02° |
| **POS** |  |  |  |
| SD Horizontal | 0.1 m | 0.1 m | 0.1 m |
| SD Vertical | 0.1 m | 0.1 m | 0.1 m |


The results show the lowest horizontal uncertainty for data collected using EM2040 (THU = 1.66 to 4.94 m), while those
collected with EM304 present the highest uncertainty (THU = 20.03 m) (Table 7). The lowest vertical uncertainty was obtained
for EM712 (TVU= 1.29 m), whilst the highest for EM2040 (TVU = 4.77 m). Uncertainties of the collected data for all the
systems fell within the limits of the IHO Standards for a specific water depth or Order number.








Table 7. Mean horizontal and vertical uncertainties of bathymetric data collected using different multibeam systems, and the accepted IHO
371                      error limits, which shows that the data collected are within the IHO standards.


| THU (m) | TVU (m) | IHO S44 6th Order Error Limit | IHO S44 6th Order No. |
|---------|---------|-------------------------------|------------------------|
| EM2040  | 1.66 - 4.94 | 0.88 - 4.77 | 2.57 |
| EM712   | 8.98    | 1.29 | 5.28 |
| EM304   | 20.03   | 3.67 | 22.80 |



The uncertainty values calculated for JammeGaia22 survey data testify that the seafloor map of the Gulf of Naples obtained
with the innovative technologies installed on board the R/V Gaia Blu represents a product of high quality. This new dataset
will serve as a crucial baseline for future in-depth analysis of the geomorphology of the area, favoring the identification of
seabed features at unprecedented resolution.
A significant improvement in the resolution of the data appears evident when comparing the morphology of the Ischia debris
avalanche from DTM at 20 m horizontal resolution generated from the ancient and modern datasets. The newly acquired
dataset shows better coverage and less noise than the 2001 dataset (Figure 9). The blocks of the landslide deposit can be also
clearly identified in the new dataset whilst the identification is not obvious for some areas in the 2001 dataset.
To test if this increase in the resolution has an impact on geomorphological indices derived from the bathymetry, we calculated
the fine-scale BPI from the 20 m-resolution DTMs (2001 and the JammeGaia22 surveys) using the same parameters for both
the datasets, reported in Table 5. The results show a much higher noise level for the 2001 DTM with respect to the
JammeGaia22 dataset (Figure 10). The noise was higher especially at the overlap among the swaths on the western part of the
dataset, and the central beams of the swath in the central part of the data, where most of the landslide blocks occur. Such blocks
are better detected and isolated through BPIs in 2022 DTM, rather than in 2001 DTM.

**4.4 Comparison to previous data**
The area for this study was selected not only for its intriguing dynamic, tectonic and volcanic activity, benthic boundary
processes and seafloor biodiversity, and widespread human impacts of various origins. An additional reason was offered by
the opportunity to compare the newly acquired data with a previous high-standard multibeam study of the area. In fact, this
area has been already mapped since the late '90s with state of the art (for that time) instrumentation and presented in extremely
accurate and imaginative 3D views (D'Argenio et al., 2004; de Alteriis et al., 2010; Passaro et al., 2014; Sacchi et al., 2014;
Budillon et al., 2016; Paoletti et al., 2016; Passaro et al., 2016a, 2016b; Di Martino et al., 2021; Aiello and Sacchi, 2022). The
limitation of that original database came from the need to acquire the data in a succession of surveys spanning several years
and using instruments with rather variable resolutions. Nevertheless, also thanks to the extreme accuracy of the data processing
performed at that time, this 20-year-old database provided an excellent basis for comparison with the newly acquired, more
homogenous, database. Of course, the comparison cannot be pushed to the highest resolution offered by the modern
instruments on Gaia Blu but, even on lower resolution, the comparison among 20 m grids from the two data sets can be
extremely valuable.

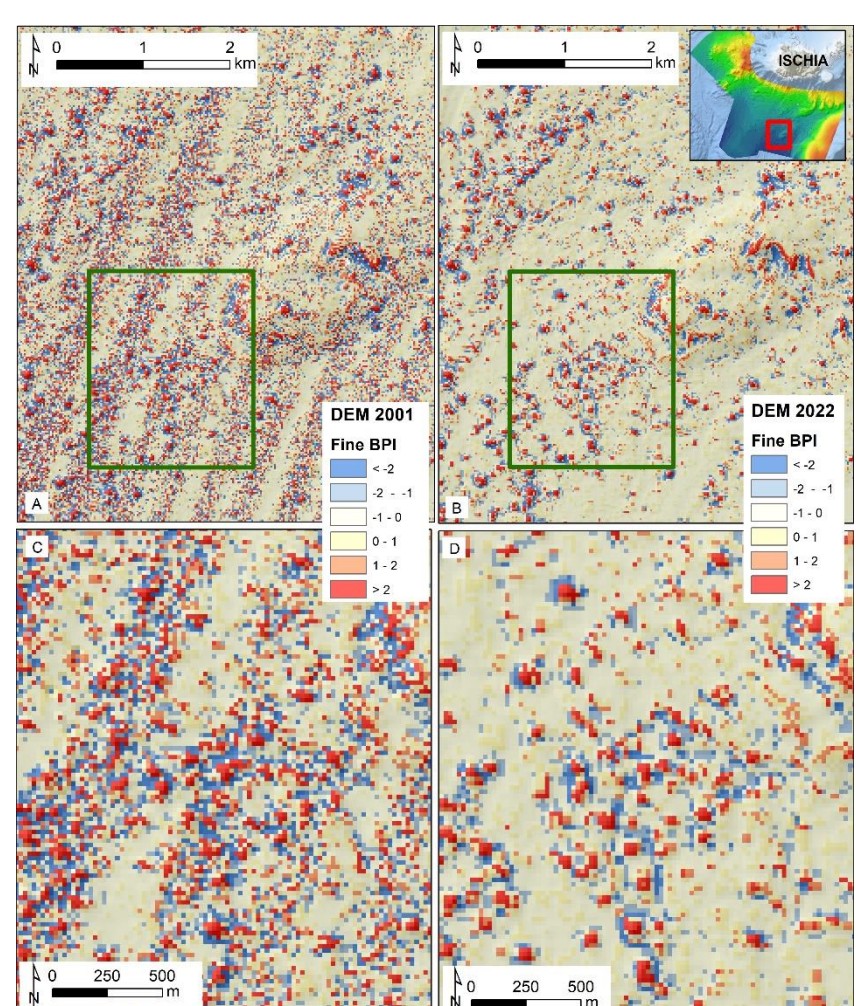



Figure 10. Fine-scale BPI calculated on the 2001 DTM (A) and JammeGaia22 DTM (B) for the area of the Ischia debris avalanche;
noticeably, the 2001 dataset is very noisy. Detail of the blocks accumulation for 2001 DTM (C) and JammeGaia22 DTM (D): despite both
datasets have same spatial resolution (20 m), the newly acquired dataset allows to better discriminate and map blocks.



5. **Data availability**
All datasets, products and web services are managed through the ISMAR Marine Spatial Data Infrastructure – MSDI (Foglini
& Grande 2023) and follow the ISMAR-CNR Data policy (https://doi.org/10.26383/CNR-ISMAR.2023.6). Bathymetric
datasets gathered by the MBES in the format GSF (generic sensor format), and bathymetric and backscatter surfaces (GeoTIFF)
are shared in the Marine Geoscience Data System (MGDS) (Table 8).
Data are also available as Web Map Services (WMS), that are interoperable with other infrastructures and permit the integration
of the spatial data in other geoportals or directly in a desktop environment (e.g., QGIS, ArcMap). Data are free accessible
thought two main interfaces: the metadata catalogue and the WebGIS.
The CNR-ISMAR GeoNetwork metadata catalogue (https://www.ismar.cnr.it/en/infrastructures/information-resources/geo-
portals/#2) allows users to find the JammeGaia22 products (refer to Table 8 for direct links to products),containing information
about access and use policy, link to download the data, how to cite the data, DOI, and links to external repositories (such as
EMODnet and MGDS).  The WebGIS (https://www.ismar.cnr.it/en/infrastructures/information-resources/geo-portals/#1)
publishes survey areas, multibeam navigation lines, bathymetric surfaces and backscatter mosaics. Users can navigate the map
to the JammeGaia22 survey area, explore the layer list and open the geophysical data and products. By clicking on spatial
objects on the map, users can access the related information, such as the download link.
Table 8. Products of the JammeGaia22 oceanographic cruise with relative link.

| Product | Typology | Depth range | Spatial resolution | Format | Link CNR-ISMAR Catalog | DOIs |
|---|---|---|---|---|---|---|
| **Survey JAMME GAIA 2022** | Cruise report | - | - | PDF | http://libeccio.bo.ismar.cnr.it:8080/geonetwork/srv/eng/catalog.search#/metadata/6cd1080c-f41f-4c9d-907b-297d25f554e5 | Foglini, et al., 2024a, https://doi.org/10.26383/CNR-ISMAR.2024.4 |
| **JG22_SwathLines_EM2040** | MBES processed lines | - | - | GSF | http://libeccio.bo.ismar.cnr.it:8080/geonetwork/srv/eng/catalog.search#/metadata/6213658d-ca9a-4e40-af07-e4f7b329203a | Foglini, 2024a http://dx.doi.org/10.60521/331589 |
| **JG22_SwathLines_EM712** | MBES processed lines | - | - | GSF | http://libeccio.bo.ismar.cnr.it:8080/geonetwork/srv/eng/catalog.search#/metadata/62136 | Foglini 2024b |



| | | | | | 58d-ca9a-4e40-af07-e4f7b329203a | http://dx.doi.org/10.60521/331587 |
|---|---|---|---|---|---|---|
| **JG22_SwathLines_EM304** | MBES processed lines | - | - | GSF | http://libeccio.bo.ismar.cnr.it:8080/geonetwork/srv/eng/catalog.search#/metadata/6213658d-ca9a-4e40-af07-e4f7b329203a | Foglini 2024c, http://dx.doi.org/10.60521/331584 |
| **JG22_50_120_2m** | Bathymetric surface | 50-120 m | 2 m | ASCII GeoTIFF ESRI_grid | http://libeccio.bo.ismar.cnr.it:8080/geonetwork/srv/eng/catalog.search#/metadata/927334e6-021a-4eed-a0a6-f209df3b17ad | |
| **JG22_100_200_5m** | Bathymetric surface | 100 -200 m | 5 m | ASCII GeoTIF ESRI_grid | http://libeccio.bo.ismar.cnr.it:8080/geonetwork/srv/eng/catalog.search#/metadata/5e384b50-ea4d-4e68-b023-d5b64ebd5ed8 | |
| **JG22_180_500_10m** | Bathymetric surface | 180-500 m | 10 m | ASCII GeoTIFF ESRI_grid | http://libeccio.bo.ismar.cnr.it:8080/geonetwork/srv/eng/catalog.search#/metadata/e956cee4-ba1c-41b7-932b-4031932c9a9d | Foglini et al. 2024b, http://dx.doi.org/10.60521/331667 |
| **JG22_480_700_15m** | Bathymetric surface | 480-700 m | 15 m | ASCII GeoTIFF ESRI_grid | http://libeccio.bo.ismar.cnr.it:8080/geonetwork/srv/eng/catalog.search#/metadata/5124f1d9-982c-4996-8333-298eb62e5c73 | |
| **JG22_680_1000_20m** | Bathymetric surface | 680-1000 m | 20 m | ASCII GeoTIFF ESRI_grid | http://libeccio.bo.ismar.cnr.it:8080/geonetwork/srv/eng/catalog.search#/metadata/21481 | |





| | | | | | |
|---|---|---|---|---|---|
| | | | | | 1a5-1700-413f-9b3f-95d2ddd29996 |
| **JG22_980_1300_30m** | Bathymetric surface | 980-1300 m | 30 m | ASCII GeoTIFF ESRI_grid | http://libeccio.bo.ismar.cnr.it:8080/geonetwork/srv/eng/catalog.search#/metadata/a43cf1d4-abc6-43e4-9f66-fac08827c5dd | |
| **JG22_1280_2120_40m** | Bathymetric surface | 1280-2120 m | 40 m | ASCII GeoTIFF ESRI_grid | http://libeccio.bo.ismar.cnr.it:8080/geonetwork/srv/eng/catalog.search#/metadata/96388cc5-2c58-4ba3-9816-7231c69d96e8 | |
| **JG22_2040_5m** | Backscatter mosaic from EM2040 | - | 5 m | ASCII GeoTIFF ESRI_grid | http://libeccio.bo.ismar.cnr.it:8080/geonetwork/srv/eng/catalog.search#/metadata/6ec52054-ac6c-46e6-966b-8a88d1cf4351 | |
| **JG22_712_10m** | Backscatter mosaic from EM712 | - | 10 m | ASCII GeoTIFF ESRI_grid | http://libeccio.bo.ismar.cnr.it:8080/geonetwork/srv/eng/catalog.search#/metadata/d4c1635f-69f2-4ebc-9174-d2a9d60a1e58 | Foglini et al. 2024c, http://dx.doi.org/10.60521/331668 |
| **JG22_304_30m** | Backscatter mosaic from EM304 | - | 30 m | ASCII GeoTIFF ESRI_grid | http://libeccio.bo.ismar.cnr.it:8080/geonetwork/srv/eng/catalog.search#/metadata/94f61db5-c186-48a6-b82b-7d9685c2a541 | |


**6. Conclusions**

The JammeGaia22 cruise led to the creation of a multi-resolution DTM and backscatter mosaic for the Gulf of Naples, by
using three different state-of-the-art MBESs. The dataset has been obtained through a reproducible processing workflow and



corresponds to a major upgrade of a pre-existing bathymetry of the area. The vertical and positioning uncertainties of the
bathymetric data fall within the IHO standards and satisfy Order 1b for EM2040 and Order 2 for EM712 and EM304.
The newly acquired multi beam maps reveal submerged morphologies at a scale and resolution never achieved before for the
study area, allowing for a wide range of local and regional studies, spanning from geological and geomorphological research
to marine habitat mapping and sea-floor monitoring. Furthermore, these high-resolution bathymetry and backscatter datasets
can be useful for Maritime spatial Planning and for designing innovative conservation strategies.
The new data base is released to the community as a benchmark reference against which future sea-floor changes can be
quantified and ascribed to either the activity of subaqueous volcanic apparatuses, in particular in the vicinity of the Flegraean
Field, the flux of density flows along major conduits like Cuma Channel, and Magnaghi and Dohrn Canyons, slope instability
leading to mass-transport deposits or sand splays at the mouth of slope gullies. Large scale bedforms are particularly developed
in regions flow rearrangement like in a bend of Cuma Channel, west of Ischia Island, or in the area of possible cyclic steps, on
the slope south of Ischia. Backscatter data help recognizing areas of potential occurrence of white coral colonies, a key element
of the Mediterranean biodiversity richness. Finally, both bathymetric and backscatter data help define the areas most impacted
by fish trawling, smoothing and remoulding the seafloor, illegal dumping and diffused littering.
**7. Author contribution**
FF: Supervisor, data collection and processing, conceptualisation, and writing; MR: Supervisor, data collection,
conceptualisation; RT: Supervisor, data collection and processing; GC, DG: data collection, data processing, first draft writing;
VG, MP: data management, data processing, first draft writing; LP, CP, FB, FM, MC, MS, ML, PM data collection and review;
GD, SI, ANT, AP, AM, AR data collection and processing; FT: Supervisor and review.
**8. Competing interests**
The contact author has declared that none of the authors has any competing interests.
**9. Acknowledgements**
We thank captain, crew, and scientific staff of R/V Gaia Blu for their skilful and efficient cooperation during operations at sea.
This is ISMAR-Bologna scientific contribution no. 2088.



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
