# Peer review of "A new multi-grid bathymetric dataset of the Gulf of Naples (Italy) from"

_Earth System Science Data, 2024_

## Author Response (AR1)

Giorgio Castellan
Institute of Marine Sciences
National Research Council of Italy
Bologna (Italy)

Email: giorgio.castellan@cnr.it
Phone: +39-051- 6398936

02/10/2024

**SUBJECT: Response to Reviewers on Article "*A new multi-grid bathymetric dataset of the Gulf of Naples (Italy) from complementary multi-beam echosounders*"**

Dear Editor,

Please find below the point-to-point reply to the comments made by reviewers on the Article "A new multi-grid bathymetric dataset of the Gulf of Naples (Italy) from complementary multi-beam echosounders".

We thank the reviewers for the overall positive evaluation of the article and their comments to improve the quality of the manuscript. We have addressed all major points raised by the reviewers.

Sincerely,

Giorgio Castellan

on behalf of all authors

| **Venezia** | **Bologna** | **Lerici** | **Napoli** | **Roma** | **Trieste** |
|---|---|---|---|---|---|
| Tesa 104 - Arsenale, | Area della Ricerca | Forte Santa Teresa, | Calata Porta Di Massa | Area della Ricerca | Area Science Park |
| Castello 2737/F | di Bologna – | Pozzuolo di Lerici | Porto Di Napoli 80 | di Roma 2 - Tor Vergata | Basovizza - Edificio Q2 |
| 30122 - Venezia, IT | Via Gobetti 101 | 19032 - La Spezia, IT | 80133 - Napoli, IT | Via del Fosso del Cavaliere 100 | Strada Statale 14, km 163.5 |
| +39 041 2407911 | 40129 - Bologna, IT | +39 0187 1788900 | +39 081 5423802 | 00133 - Roma, IT | 34149 - Trieste, IT |
| protocollo.ismar@pec.cnr.it | +39 051 639 8891 | | | +39 06 45488634 | +39 040 3756872 |
| www.ismar.cnr.it | | | | | |

**Reviewer 1**

**We truly thank the reviewer for the pertinent comments and inputs to improve the manuscript quality, which has been modified following the points raised by the reviewer. Here below is our point-by-point response to the reviewer comments. Please, notice that italics are used for reviewer comments whilst bold is used for our responses.**

**We hope that with our effort we satisfyingly addressed the major concerns of the reviewer.**

*This paper describes high resolution bathymetric data acquired in the Gulf of Naples. This paper participates, at its local scale, to this effort of ocean mapping and the global effort to make this data available for multiple usages. After having said this, the paper essentially reports classical data processing, and classical strategy to distribute the data on several national and international data portals, hence the rating above.*

*Following, you will find detailed points all along the text:*

*- introduction:*

*l41 I would add after "benthic habitats" "to cite a few applications"*

**Modified**

*l65: strange phrasing but ok: "with an average vertical resolution of less than 0.25 % of the water depth" I would delete "average"*

**Modified**

*l75: Please indicate why you think EMODnet Bathymetry does not satisfy FAIRness*

**We do not, indeed our statement referred to local datasets. We changed the sentence to clarify our point.**

*l90: "near real time" what is your definition of near-real time (minutes, hours,), while surveying. I would not pretend for "near real time" but would speak about "acceleration/accelerated process". Note that because of tide correction/vertical referencing, the process is asynchronous.*

**We changed the sentence according to the comment.**

*3.Material and methods*

*l153: "A software tool". You might want to indicate the name of the software. Rigorously speaking, if your pretend FAIRness of your dataset you need to detail every step, including the name of the software you've used.*

**We specified the tool used for SVP extension.**

*l165-166 I would also add to this sentence "and poor seafloor detection" (phase detection for outer beams is less robust than amplitude detection for the center beams)*

**Modified.**

*The reviewer would like if any backscatter calibration has been done considering multiple surveys with multiple*

**Venezia**
Tesa 104 - Arsenale,
Castello 2737/F
30122 - Venezia, IT
+39 041 2407911
protocollo.ismar@pec.cnr.it
www.ismar.cnr.it

**Bologna**
Area della Ricerca
di Bologna –
Via Gobetti 101
40129 - Bologna, IT
+39 051 639 8891

**Lerici**
Forte Santa Teresa,
Pozzuolo di Lerici
19032 - La Spezia, IT
+39 0187 1788900

**Napoli**
Calata Porta Di Massa
Porto Di Napoli 80
80133 - Napoli, IT
+39 081 5423802

**Roma**
Area della Ricerca
di Roma 2 - Tor Vergata
Via del Fosso del Cavaliere 100
00133 - Roma, IT
+39 06 45488634

**Trieste**
Area Science Park
Basovizza - Edificio Q2
Strada Statale 14, km 163.5
34149 - Trieste, IT
+39 040 3756872

*sounding system. By calibration, we mean a procedure such as by Eleftherakis et al. (see https://link.springer.com/article/10.1007/s11001-018-9348-5) or Misiuk et al (doi: 10.3390/rs12040601)*

**No calibration was carried out, neither during the acquisition nor in post-processing. The method proposed by Eleftherakis et al (2018) was not feasible to be applied for the following reasons:**

- **the single beam is mounted under the vessel keel and cannot be moved to change the inclination and measure absolute backscatter as suggested by Eleftherakis et al (2018), and**
- **the procedure required more time at sea specifically dedicated and it was not possible at the time of the survey.**

**Misiuk et al (2020) proposed models to apply the "bulk shift approach" on mosaics acquired with**

- **same MBES but operated at different frequencies, or**
- **same MBES operated at the same frequencies in different time frames to simulate repeated surveys.**

**Neither of these is the case of the Jammegaia2022 survey, but for sure backscatter calibration will be faced and examined in depth in a successive dedicated work, enriched and ground-truthed by seabed samples and images that were not collected during the Jammegaia2022 survey.**

*Figure 2 Box "Apply "strong spline filter""-> please document; "Apply backscatter correction" -> how those are being done, please document also; "Correzione del beam pattern" -> Switch to full English*

**We modified the Figure. The steps are described in paragraphs 3.2.1 and 3.2.2, now specified in the caption.**

*l182: I would add, at the end of the sentence " from a previous swath".*

**Added.**

*Related to the previous point: Have you cross-checked also with previous survey in the area (even from another institute). Ultimately, if you want to strongly document the robustness of your data you should cross check with known Ground Point Control - geologically speaking fixed seabed features)*

**Yes, we cross-checked with the survey carried out by the CNR in 2001 and used it for the comparison described in this paper.**

*l188 Nevertheless, we typically applied the 'Quick Search' ... It seems to me very specific. It does not seem to me to bring useful information for the reader who might want to reprocess your data (in the context of FAIRness)*

**We removed the sentence.**

*Table 3 "Insonified area" either in sq. m or $m^2$. Please add, if possible, indicating the theoretical number of soundings per grid node*

**We included information on the soundings, in terms of mean number of soundings per cell of the grid. We report the results in the table below and updated Table 3 in the text.**

*l203 "to improve the spatial accuracy" -> actually what your are doing here is simply georeferencing the backscatter value, please use this term*

**Modified.**

| **Venezia** | **Bologna** | **Lerici** | **Napoli** | **Roma** | **Trieste** |
|---|---|---|---|---|---|
| Tesa 104 - Arsenale, | Area della Ricerca | Forte Santa Teresa, | Calata Porta Di Massa | Area della Ricerca | Area Science Park |
| Castello 2737/F | di Bologna – | Pozzuolo di Lerici | Porto Di Napoli 80 | di Roma 2 - Tor Vergata | Basovizza - Edificio Q2 |
| 30122 - Venezia, IT | Via Gobetti 101 | 19032 - La Spezia, IT | 80133 - Napoli, IT | Via del Fosso del Cavaliere 100 | Strada Statale 14, km 163.5 |
| +39 041 2407911 | 40129 - Bologna, IT | +39 0187 1788900 | +39 081 5423802 | 00133 - Roma, IT | 34149 - Trieste, IT |
| protocollo.ismar@pec.cnr.it | +39 051 639 8891 | | | +39 06 45488634 | +39 040 3756872 |
| www.ismar.cnr.it | | | | | |

*l204 the reference grid -> the bathymetric grid*

**Modified.**

*l206 "spot" -> "position"*

**Modified.**

*l207 "correct scaling" please explain*

**We modified the sentence.**

*l216/217please associate appropriate reference(s) for the BPI. Likewise for rugosity as there are multiple understanding of what rugosity is (note the scale dependency)*

**We kindly report that References for BPI are reported in the next paragraph. Now we also included descriptions and references for the terrain ruggedness.**

*l219 No need to add in the text developed by Dr Tim Le Bas, considering you provide an explicit citation, although I truly acknowledge Dr Le Bas work.*

**Modified.**

*l222 to be rigorous you should provide the underlying algorithm used to compute the slope*

**Modified.**

*l225: It is not clear to me at all how BPI is a second derivative, the BPI definition does not specify that it is derived from the slope (it is derived from the difference of focal calculations of first order bathymetry/topography at 2 scales)*

**Modified.**

*l229 "denoting the reliefs" -> "denoting positive reliefs"*

**Modified.**

*Figure 3: 2 vertical exaggeration -> " x 2 vertical exaggeration"*

**Modified.**

*l246: "hugging" the seafloor: please verify this is an appropriate term (scientifically accepted, personally not sure)*

**Modified.**

*l257: showing differences. Please indicate explicitly what are you differentiating.*

**Modified.**

*l271 DPI -> BPI*

**Modified.**

**Venezia**
Tesa 104 - Arsenale,
Castello 2737/F
30122 - Venezia, IT
+39 041 2407911
protocollo.ismar@pec.cnr.it
www.ismar.cnr.it

**Bologna**
Area della Ricerca
di Bologna –
Via Gobetti 101
40129 - Bologna, IT
+39 051 639 8891

**Lerici**
Forte Santa Teresa,
Pozzuolo di Lerici
19032 - La Spezia, IT
+39 0187 1788900

**Napoli**
Calata Porta Di Massa
Porto Di Napoli 80
80133 - Napoli, IT
+39 081 5423802

**Roma**
Area della Ricerca
di Roma 2 - Tor Vergata
Via del Fosso del Cavaliere 100
00133 - Roma, IT
+39 06 45488634

**Trieste**
Area Science Park
Basovizza - Edificio Q2
Strada Statale 14, km 163.5
34149 - Trieste, IT
+39 040 3756872

*l280: characterize -> characterized*

**Modified.**

*l282: Peering -> Pairing*

**Modified.**

*l293: see my previous comment about the acceptance, as a scientific term, of the term "hug".*

**Modified.**

*l333: Please explain how backscatter grids are cross calibrated. Also note more globally that you should not state in the title multi-resolution backscatter mosaic. I would personally replace all the occurrences of multi-resolution by something like mosaic with multiple grid at their specific resolution. multi-resolution mosaic would mean that you have a single data structure with an adapted resolution gridding system (quadtree for example). I don't think this is the case*

**The backscatter grids were not cross-calibrated. The processing of the backscatter is described in the paper: backscatter data were extracted from .gsf files and corrected using the tools offered by Fledermaus: backscatter corrections, beam pattern correction, and angle-varying gain (AVG) correction. Also, the backscatter mosaics were georeferenced and corrected for the topography using the processed bathymetry. No calibration was carried out, neither during the acquisition nor in post-processing, because the methods described in the literature did not apply to the Jammegaia2022 survey. A dedicated work will be prepared to analyze the backscatter calibration in deep for this survey.**

*Fig 9 please provide the backscatter color palette with associated legend*

**Modified**

*l376 "innovative technologies". Although complex technologies, I would not state that MBES are innovative, as it has been here for several decades now.*

**Modified.**

*l395 "imaginative" likewise, this term does not bring much to your text*

**Modified.**

*l416 free -> freely*

**Modified.**

*At the date (25/07) when I reviewed this paper https://www.ismar.cnr.it/en/infrastructures/information-resources/geo-portals/#1 and https://www.ismar.cnr.it/en/infrastructures/information-resources/geo-portals/#2 ------> both are leading to 404 error (unFAIR)*

**We provided the new links and updated the text accordingly:**

- **ISMAR Geoportal: http://seamap-explorer.data.ismar.cnr.it:8080/mokaApp/applicazioni/ismarBoApp**
- **ISMAR Geonetwork: http://seamap-catalog.data.ismar.cnr.it:8080/geonetwork.**

**Venezia**
Tesa 104 - Arsenale,
Castello 2737/F
30122 - Venezia, IT
+39 041 2407911
protocollo.ismar@pec.cnr.it
www.ismar.cnr.it

**Bologna**
Area della Ricerca
di Bologna –
Via Gobetti 101
40129 - Bologna, IT
+39 051 639 8891

**Lerici**
Forte Santa Teresa,
Pozzuolo di Lerici
19032 - La Spezia, IT
+39 0187 1788900

**Napoli**
Calata Porta Di Massa
Porto Di Napoli 80
80133 - Napoli, IT
+39 081 5423802

**Roma**
Area della Ricerca
di Roma 2 - Tor Vergata
Via del Fosso del Cavaliere 100
00133 - Roma, IT
+39 06 45488634

**Trieste**
Area Science Park
Basovizza - Edificio Q2
Strada Statale 14, km 163.5
34149 - Trieste, IT
+39 040 3756872

*l428: see my previous comment about multi-resolution vs grids at multiple resolutions*

**Modified.**

*l431: Be careful that IHO standard S44 are more than TVU and THU, which you have discussed in your paper. If you want to be fully S44 compliant you should also swath overlapping, detection capabilities. You might also want to have a survey with appropriate redundancy.*

**The bathymetric coverage is 100% (= full bathymetric coverage) ensured through the compromise between coverage overlapping between adjacent swaths (20% in the Jammegaia2020 survey) and instrumental swath coverage. This is clarified in the Materials and Methods section 3.1 Multi-beam data acquisition.**

**According to table "Minimum Bathymetry Standards for Safety of Navigation Hydrographic Surveys" p.28 of the S-44 (https://iho.int/uploads/user/pubs/standards/s-44/S-44_Edition_6.1.0.pdf), detection capability is not required by the S-44 IHO Standard Edition 6.1 for the Order 2 and Order 1b (see section 7.3, Table 1 at page 18).**

**The redundancy and the quality of data ensure the production of high-resolution bathymetry with an optimal quality of data for the scopes of the survey.**

*l435: "MSP and designing innovative conservation strategies" There is much more applications than these cited here.*

**Modified the text accordingly.**

*l442-l443. You should remove the last line as you have not really provided this example earlier in your text. In a conclusion, you should not bring new scientific elements. However, providing a section in your paper that describes through visual examples, would be nice.*

**Modified accordingly**

*References:*

*Please verify Dolan MFJ Calculation ... l 539. It seems that the reference is incomplete.*

**Modified**

*l620. Moussat et al. Try not to use shortenings. -> Compte rendu de l'Académ ie des Sciences, Série 2, Mécanique, physique, chimie, sciences de l'univers, sciences de la terre*

**Modified.**

**Venezia**
Tesa 104 - Arsenale,
Castello 2737/F
30122 - Venezia, IT
+39 041 2407911
protocollo.ismar@pec.cnr.it
www.ismar.cnr.it

**Bologna**
Area della Ricerca
di Bologna –
Via Gobetti 101
40129 - Bologna, IT
+39 051 639 8891

**Lerici**
Forte Santa Teresa,
Pozzuolo di Lerici
19032 - La Spezia, IT
+39 0187 1788900

**Napoli**
Calata Porta Di Massa
Porto Di Napoli 80
80133 - Napoli, IT
+39 081 5423802

**Roma**
Area della Ricerca
di Roma 2 - Tor Vergata
Via del Fosso del Cavaliere 100
00133 - Roma, IT
+39 06 45488634

**Trieste**
Area Science Park
Basovizza - Edificio Q2
Strada Statale 14, km 163.5
34149 - Trieste, IT
+39 040 3756872

**Reviewer 2**

We thank the reviewer for the careful reading of the manuscript and their constructive remarks. We have taken the comments on board to improve and clarify the manuscript. Please find below a detailed point-by-point response to all comments (reviewers' comments in italics, our replies in bold).

*Very interesting paper covering the whole process from collection at sea (including technical aspects) to creation of useful products for scientific use, many thanks to give me the occasion to read it!*

*As a hydrographer, I focused on the technical aspect and quality evaluation of the data collection. Several questions are raised because I'm interested to know the answers but also because I think it needs to be clarified in the paper for a better understanding by the reader.*

*line 78: I suggest to introduce the name of the survey campaign her in order no to introduce confusion in the following lines. Moreover, I also suggest to present this part in a more structured way in order to present all the sections (and potentially the sub-sections) to give to the reader a better introduction to the paper (and give him the will to continue!).*

**Modified accordingly**

*line 151: I understand the SV was permanently collected near the transducers. This information was not used to decide when it was necessary to collect a new SVP (I see the SVPs were collected twice a day) ? During Shom's surveys, the policy is to collect a new SVP (with Sippican probe at least) when the SV measure is greater than a certain value regarding the one at the last SVP (typically > 2 m/s). And after, we apply the "nearest in time" profiles to the bathymetric data.*

**Modified.**

*line 161 : Did you consider the change of transducers' draft during the survey (ie by applying a linear regression between the draft measured at the beginning and at the end of the survey) ? It doesn't change the result a lot, but if applied it's good to be mentionned.*

**The transducer's draft was not modified since no significant changes were noted during the survey.**

*line 165 : what about the sea state during the survey ?*

**We added information on sea conditions.**

*line 171 : I understand the collected lines were processed in almost realtime on board in order to make the DTMs available as soon as possible. Does it mean that DTMs were processed line by line and compiled progressively (if it's the case, how did you deconflict in the common areas ?) or did you make global DTMs for each resolution with all the processed lines at the end of the survey ?*

**We clarified this point in the section.**

*line 195 : table 3 : I do not understand the column "Insonified area". What represent the values ? What is the unit ? Wouldn't it be better to use square meters ? Moreover, the best parameter to define the grid resolution is the number of measurements located in the cell (at least 10 by cell is a good value).*

**The insonified area is expressed in m$^2$ and we modified the text in the table accordingly.**

**Venezia**
Tesa 104 - Arsenale,
Castello 2737/F
30122 - Venezia, IT
+39 041 2407911
protocollo.ismar@pec.cnr.it
www.ismar.cnr.it

**Bologna**
Area della Ricerca
di Bologna –
Via Gobetti 101
40129 - Bologna, IT
+39 051 639 8891

**Lerici**
Forte Santa Teresa,
Pozzuolo di Lerici
19032 - La Spezia, IT
+39 0187 1788900

**Napoli**
Calata Porta Di Massa
Porto Di Napoli 80
80133 - Napoli, IT
+39 081 5423802

**Roma**
Area della Ricerca
di Roma 2 - Tor Vergata
Via del Fosso del Cavaliere 100
00133 - Roma, IT
+39 06 45488634

**Trieste**
Area Science Park
Basovizza - Edificio Q2
Strada Statale 14, km 163.5
34149 - Trieste, IT
+39 040 3756872

**We included information on the mean number of soundings per cell of the grid. We report the results in Table 3 in the text.**

*line 351 : the whole §4.3 : I do not understand the Table 7. Be careful when you are refering to the IHO S-44. S-44 standard is not only about H/V incertainties, but also about data coverage and feature detection as this standard is first of all dedicated to safety of navigation. Please, have a look at the table "Minimum Bathymetry Standards for Safety of Navigation Hydrographic Surveys" p.28 of the S-44 (https://iho.int/uploads/user/pubs/standards/s-44/S-44_Edition_6.1.0.pdf). The first line gives the classification of the survey. In your case, underkeel clearance is not critical so order 1a or better is not an objective. You can apply to order 1b or 2 and just have to justify the respect of THU and TVU. The better way to do that is to evaluate these uncertainties and modeling them with formulas like a+b\*Depth (depending on the MBES) and compare it to the IHO ones (1+0,023\*Depth and 0,5 + 0,013\*Depth). I think the section should be reformulated in this way.*

**We thank the reviewer for the pertinent comment. A format error occurred in Tab. 7 and we now have fixed including also TVU and THU for Order 2. A more explicit detail on data coverage is now included in section 3.1. In line with S-44, information on "feature detection" is not included since not required for Orders 1 and 2, according to table "Minimum Bathymetry Standards for Safety of Navigation Hydrographic Surveys" p.28 of the S-44.**

*line 401 : Better to write "Gaia Blu" in the same way in all the paper (ie. In capital letters).*

**The name has been changed to "Gaia Blu" in the whole text since it's the name used also in the DOI links.**

*line 431 : the IHO S-44 orders reached are given here, but not clearly in the Data quality dedicated section.*

**The data quality section now includes better discuss the IHO S-44 Orders.**

*These are my quick comments on this paper which is anyway a very interesting one on a survey contributing to the global effort of ocean mapping !*

*Gaël*

**Venezia**
Tesa 104 - Arsenale,
Castello 2737/F
30122 - Venezia, IT
+39 041 2407911
protocollo.ismar@pec.cnr.it
www.ismar.cnr.it

**Bologna**
Area della Ricerca
di Bologna –
Via Gobetti 101
40129 - Bologna, IT
+39 051 639 8891

**Lerici**
Forte Santa Teresa,
Pozzuolo di Lerici
19032 - La Spezia, IT
+39 0187 1788900

**Napoli**
Calata Porta Di Massa
Porto Di Napoli 80
80133 - Napoli, IT
+39 081 5423802

**Roma**
Area della Ricerca
di Roma 2 - Tor Vergata
Via del Fosso del Cavaliere 100
00133 - Roma, IT
+39 06 45488634

**Trieste**
Area Science Park
Basovizza - Edificio Q2
Strada Statale 14, km 163.5
34149 - Trieste, IT
+39 040 3756872